# Molecular Classification of Gastric Cancer among Alaska Native People

**DOI:** 10.3390/cancers12010198

**Published:** 2020-01-13

**Authors:** Holly A. Martinson, Dominic Mallari, Christine Richter, Tsung-Teh Wu, James Tiesinga, Steven R. Alberts, Matthew J. Olnes

**Affiliations:** 1WWAMI School of Medical Education, University of Alaska Anchorage, 3211 Providence Drive, Anchorage, AK 99508, USA; 2Department of Chemistry, University of Alaska Anchorage, 3211 Providence Drive, Anchorage, AK 99508, USA; dmmallari@gmail.com; 3Department of Biological Sciences, University of Alaska Anchorage, 3211 Providence Drive, Anchorage, AK 99508, USA; christine.richter15@gmail.com; 4Department of Laboratory Medicine and Pathology, Mayo Clinic Cancer Center, 200 First Street SW, Rochester, MN 55905, USA; Wu.TsungTeh@mayo.edu; 5Department of Pathology and Clinical Laboratory, Alaska Native Medical Center, 4315 Diplomacy Drive, Anchorage, AK 99508, USA; jjtiesinga@anthc.org; 6Medical Oncology, Mayo Clinic Cancer Center, 200 First Street SW, Rochester, MN 55905, USA; alberts.steven@mayo.edu; 7Oncology and Hematology, Alaska Native Medical Center, 4315 Diplomacy Drive, Anchorage, AK 99508, USA; mjolnes@anthc.org

**Keywords:** Alaska Native people, signet ring, diffuse, *Helicobacter pylori*, Epstein-Barr virus, COX-2, MUC1, *TP53*, PD-L1, Mismatch repair deficiency

## Abstract

Gastric cancer is an aggressive and heterogeneous malignancy that often varies in presentation and disease among racial and ethnic groups. The Alaska Native (AN) people have the highest incidence and mortality rates of gastric cancer in North America. This study examines molecular markers in solid tumor samples from eighty-five AN gastric adenocarcinoma patients using next-generation sequencing, immunohistochemistry, and in situ hybridization analysis. AN patients have a low mutation burden with fewer somatic gene mutations in their tumors compared to other populations, with the most common mutation being *TP53*. Epstein-Barr virus (EBV) was associated with 20% of AN gastric cancers, which is higher than the world average of 10%. The inflammation marker, cyclooxygenase-2 (COX-2), is highly expressed in patients with the lowest survival rates. Mismatch repair deficiency was present in 10% of AN patients and was associated with patients who were female, 50 years or older, gene mutations, and tumors in the distal stomach. Program death-ligand 1 (PD-L1) was expressed in 14% of AN patients who were more likely to have MMR deficiency, EBV-associated gastric cancers, and mutations in the *PIK3CA* gene, all of which have been linked to clinical response to PD-1 inhibitors. These studies suggest a portion of AN gastric cancer patients could be candidates for immunotherapy. Overall, this study highlights future avenues of investigation for clinical and translational studies, so that we can improve early detection and develop more effective treatments for AN patients.

## 1. Introduction

Gastric cancer is the fifth most common cancer and the third leading cause of cancer-related death worldwide [1]. The prevalence of gastric cancer varies geographically, with the majority of cases occurring in the developing countries and high-risk areas such as in Central and South America, Eastern Europe, and East Asia countries [1,2]. The majority of gastric cancer cases are adenocarcinomas, which are commonly subdivided into intestinal and diffuse types, according to Lauren classification [3]. Gastric cancer patients are often diagnosed at advanced stages because of the lack of screening strategies, leaving most patients with limited treatment options. Thus, there is an urgent need to better understand the pathogenesis of gastric cancer and to identify early detection and biomarkers that can be utilized to identify more effective and less toxic therapeutic strategies. 

The pathogenesis of gastric cancer is commonly associated with infectious agents such as *Helicobacter pylori* (*H. pylori*) [4] and Epstein-Barr virus (EBV) [5]. Persistent inflammation during *H. pylori* or EBV infection can generate changes in the gut microenvironment where inflammatory mediators activate signaling pathways that lead to gastric tumorigenesis [6,7]. *H. pylori*, in particular, has been associated with gastric cancers present in the non-cardia region of the stomach, whereas EBV associated cancers (EBVaGC) are associated with tumors located throughout the stomach [4,8]. The incidence of *H. pylori* and EBVaGC are variable across high-risk gastric cancer areas worldwide, with *H. pylori* highest in East Asia, EBV highest in the United States, and both *H. pylori* and EBV occurring more frequently in South America [9]. Other environmental factors, such as a diet high in salt and smoking, have also been shown to be significant risk factors for gastric cancer [10]. In addition to environmental agents, host genetic factors such as gene polymorphisms and hereditary cancer syndromes increase a patient’s risk of developing gastric cancer [11,12]. 

Recently, genomic technology and high-throughput analysis have helped evaluate the molecular and genetic makeup of gastric tumors. The landmark study in 2014 by The Cancer Genome Atlas (TCGA), molecularly classified gastric cancer into four subtypes: (i) Epstein-Barr virus (EBV) positive tumors, (ii) microsatellite instability (MSI) tumors, (iii) genomically stable (GS) tumors, and (iv) tumors with chromosome instability (CIN) [13]. Each of these molecular subtypes revealed distinct genomic features that could provide a guide to predictive and prognostic biomarkers and targeted agents for gastric cancer patients. 

The Alaska Native (AN) population has the highest incidence and mortality rates of gastric cancer among all ethnicities in the United States [14,15,16,17]. Gastric cancer etiology differs between the AN populations and other U.S. and Alaska populations [14]. AN gastric cancer patients are diagnosed at a younger age and have a higher prevalence of non-cardia tumors, diffuse subtype, and signet ring cell carcinomas [14]. The higher incidence of non-cardia gastric cancers among AN people has been associated with the high seropositivity rates of *H. pylori* among the general population [18,19,20,21]. A greater understanding of the molecular features of gastric cancers diagnosed in the AN people will help identify early detection and therapeutic biomarkers as well as potential treatment strategies that can be used to reduce the incidence and mortality rates of gastric cancer.

We report the results of our study, which aimed to molecularly profile a cohort of AN gastric cancer patients. Our study included demographic, clinical, protein and viral RNA expression, and molecular profiling data. We report patient survival by gender, stage, and Lauren histological type. This is the first study to describe the molecular characteristics of gastric cancer among the AN people.

## 2. Results

### 2.1. Patients’ Clinicopathological Information

We performed a retrospective analysis of 85 AN patients who had clinical biopsies or resection tissue samples taken for locally advanced gastric adenocarcinomas at the Alaska Native Medical Center. The demographic and clinical-pathological characteristics of patients were reviewed from hospital records and are summarized in Table 1. Because of limited patient tissue, next-generation sequencing analysis was performed on 82 patients and tissue expression analysis on 85 patients. Patients were predominantly male (61.2%), classified at stages III/IV (54%), and were of high grade (70.6%). Most tumors were located in the non-cardia regions of the stomach, body (24.7%), antrum (11.8%), pylorus (16.5%), or in overlapping regions (23.5%). Histologically, 51.8% of patients presented with diffuse-type and 29.4% with signet-ring cell presence.

### 2.2. Somatic Gene Mutations

Fifty genes commonly mutated in cancer were assessed by next-generation sequencing for 82 patient FFPE tumor samples. Single nucleotide variants (SNV) were detected in 57 patients, with 35 patients having only one mutation and 22 patients with two or more mutations (Table 2). The most common mutations were missense mutations in *Tp53* (32.9%), *PIK3CA* (15.7%), *STK11* (7.2%), KRAS (8.2%), and *PTEN* (7.2%) (Table 3). Patients diagnosed with the intestinal subtype were significantly more likely to have a mutation (61% intestinal vs. 38% diffuse, *p* = 0.001). Well-differentiated (low grade) tumors were more likely to have a mutation compared to poorly differentiated (high grade) tumors (92% vs. 60%, *p* = 0.003). Furthermore, patients with signet ring cell carcinoma had significantly fewer mutations than patients without signet ring cell presence (52% vs. 76%, *p* = 0.001). 

### 2.3. EBV Associated Gastric Cancers

Gastric cancer patient tissue samples were examined for latent EBV infection for EBV-encoded small ribonucleic acid 1 (EBER1) using chromogenic in situ hybridization. Overall, 19 of the 85 tumors (20.2%) examined, revealed EBER1 expression within the malignant epithelial cells (Table 4 and Figure 1). The demographic and clinical-pathological characteristics of patients with EBVaGC were reviewed from hospital records and are summarized in Table 5. EBER1 expression was present in both diffuse (11) and intestinal (9) type of gastric cancer. Expression of EBER1 was not present in non-neoplastic gastric mucosal, endothelial, or fibroblast cells. EBER1-positive benign-appearing lymphoid cells were present in 79 of the 85 patient samples (93%). EBV+ tumors were more likely to have a mutation in PIK3CA compared to EBV− tumors (38.9% vs. 10.5%, *p* = 0.006).

### 2.4. HER2 Expression

Our data indicates 9.4% of AN gastric cancer patients were HER2+ (Table 4). None of the HER2+ patients were signet-ring cell positive, 7/8 were intestinal type by Lauren classification and 6/8 were low grade, grade 2. HER2+ patients were more likely to be under the age of 50 (6/8 HER2+ patients, *p* = 0.02). 

### 2.5. MUC1 and Tumor-Associated MUC1

Fully glycosylated mucin 1 (MUC1) and hypoglycosylated MUC1 expression intensity and percent of positive tumor cells were examined by immunohistochemistry (IHC) in gastric cancer tissue sections. Patients were categorized by tumor cell expression intensity low MUC1 (Figure 2A) vs. high MUC1 (Figure 2B and Table 4). Low to moderate MUC1 expression was observed in 17.6% of patients (9 diffuse, 6 intestinal) and high MUC1 in 82% of patients (36 diffuse, 34 intestinal). Hypoglycosylated or tumor-associated MUC1 (TA-MUC1) expression in tumor cells was present in 90.6% of AN gastric cancer patients (Figure 2C,D and Table 4). High TA-MUC1 expression was observed in 23 patients, low expression in 53 patients, and no expression in 8 patients. Of the 8 patients that had no TA-MUC1 expression, 6 of the patients were diagnosed at a low stage I-II. Patients with the diffuse subtype were more likely to have high TA-MUC1 expression (37.8%, 17/45 diffuse patients) compared to intestinal subtype (15%, 6/40 intestinal patients) (*p* = 0.02). 

### 2.6. COX-2 Tumor Cell Expression and Prognosis

In our study, the intensity of staining of positive tumor cells was graded for COX-2 protein expression by IHC. Expression of COX-2 was observed in smooth muscle cells, fibroblasts, and mononuclear cells within the stroma. High COX-2 intensity was noted for the 15.3% of the patients, with four patients having no COX-2 expression (Figure 3A,B; Table 4). High COX-2 expression in tumor cells was associated with advanced stage IV (69.2% vs. 36.1%, *p* = 0.032) and poorer disease-free and overall survival (Figure 3C and Table 6). After multivariate analysis, the difference in survival between COX-2 low and high was no longer significant (HR = 1.58, 95% CI = 0.80–3.13; *p* = 0.186; Table 7). 

### 2.7. Mismatched Repair Deficiency

Mismatch repair (MMR) deficiency was assessed by IHC using antibodies directed toward three MMR proteins, namely MSH2, MSH6, and PMS2. Among the patients evaluated for MMR expression, one patient had a loss of MSH2 and MSH6, and seven patients had a loss or weak expression of PMS2 (Table 4). Patients with MMR deficiency were significantly more likely to be female (75%, *p* = 0.035) and over the age of 50 (mean age 71 vs. 60, *p* = 0.001). MMR deficiency was associated with low MUC1 expression 14/15 patients (*p* = 0.02). MMR deficient patients were also more likely to have an SNP mutation (87.5%), with 6/8 patients with more than two mutations. Of the nine MMR-deficient patients, seven of the patients had tumors in the distal (body, pyloric, and antrum) stomach location. Place of residence was also associated with MMR deficiency with 6/8 patients located in northern Alaska, which has a higher percentage of intestinal type cancers, 64.7%, compared to other regions in Alaska that range from 20–40% [14]. Patients with MMR deficiency had slightly shorter survival rates (16.5 months) but not significant compared to MMR-stable patients (24.6 months). 

### 2.8. PD-L1 Expression

PD-L1 expression on tumor cells and tumor-infiltrating stromal/immune cells were assessed by IHC. All non-neoplastic gastric epithelia were identified to be PD-L1. Percentage of PD-L1+ tumor cells ranged from 1–90% of total tumor cells and from 1–10% of total stromal/immune cells to total cells. Patients who had a combined positive score greater than 1 or 1% were classified as PD-L1 positive (17% of patients, 11 intestinal and 4 diffuse) and patients with less than 1% positive cells were classified as PD-L1 negative (83% of patients; Table 4). Immune cell PD-L1 expression was observed in four patients, of which three did not have PD-L1 tumor cell expression. PD-L1 expression was significantly correlated with loss of the mismatch repair (MMR) protein expression (MSH2, MSH6, or PMS2) by IHC (5/15 or 33% of PD-L1+ compared to 3/68 or 4.2% of PD-L1− *p* = 0.002). PD-L1 positive expression was observed in 14/48 patients with adenocarcinoma and in 1/26 patient with signet ring cell carcinoma (*p* = 0.03). EBV associated gastric cancer patients were more likely to express PD-L1, 36.8% compared to EBV negative patients, 11.8% (*p* = 0.001). We also observed PD-L1+ expression was associated with *PIK3CA* mutation (35.7% PD-L1+ vs. 14.5% PDL1−, *p* = 0.05).

## 3. Discussion

In this study, we sought to gain a deeper understanding of the molecular features of gastric cancer in the AN population. Our analysis of a targeted 50 somatic gene mutation panel confirmed 60.6% of the patients had one or more gene mutations with the *TP53* gene being the most frequently altered among AN patients, which was similar but lower in frequency, 36% when compared to other populations, 46–49% [13,22,23,24]. Other genes that were commonly mutated in AN gastric cancer patients were *PIK3CA*, *PTEN*, *KRAS*, *APC*, and *CTNN1B*, which were comparable to the TCGA study (Table 3) [13]. Furthermore, a higher frequency of mutations was observed in AN patients diagnosed with well-differentiated tumors and of the intestinal type, which was not expected. A novel *STK11* mutation, F354L, was also observed in six patients, all with intestinal type gastric cancers. The F354L variant has been previously reported in a breast, thyroid, esophageal, gastric, and neuroblastoma cancer patients [25,26,27,28], and has been suggested to be a potential biomarker for mTOR inhibitor sensitivity [26]. The incidence of genomically stable gastric cancers with a low mutation burden among AN patients was 39.4%. This estimate is higher than what has been found in other studies; however, in this study, a targeted sequencing fifty gene cancer panel was performed compared to the whole exome or whole genome sequencing in other studies. 

EBV is known to be associated with 10% of the gastric cancer cases worldwide [9]. The AN population has a 17-fold higher incidence of EBVaGC, such as nasopharyngeal carcinoma [29], which are regularly driven by EBV [30]. Therefore, we investigated the incidence of EBV in AN gastric cancer patients and observed 20% of patient tumors were positive for the latent EBV marker EBER1, indicating that the AN population exhibits a higher percentage of EBV+ gastric cancer than most reported studies, and a rate comparable to a recently reported cohort of gastric cancer patients in Peru [31]. The high incidence of EBVaGC could be the result of co-infection with EBV and *H. pylori*, which has been associated with increased risk of occurrence of gastric cancer [32]. Furthermore, EBV+ tumors were both diffuse and intestinal type and occurred more frequently in the cardia and middle stomach, similar to what has been observed in the literature [31,33]. These finding may have future potential therapeutic implications and raises the possibility that EBV may contribute to the high rate of GC in AN people.

HER2 gene amplification or protein overexpression occurs in numerous types of human cancers, including breast, gastric, colon, bladder, and lung. The immunotherapy trastuzumab was the first immunotherapy and targeted therapy option approved for gastric cancer. Among the AN people, we observed that 9.4% of gastric cancer patients were HER2+. HER2 positivity was predominately seen in patients with intestinal-type tumors, which has been previously observed in other studies [34] but was not associated with a particular tumor location in AN patients. HER2 overexpression was associated with AN patients who were under the age of 50 at the time of diagnosis, which is contrary to other studies [35]. These data suggest that further studies are warranted to investigate trastuzumab sensitivity in early-onset gastric cancer patients in high-risk populations.

Elevated MUC1 expression has been observed in gastric cancer patient tumors and is thought to play a critical role in oncogenic signaling [36]. MUC1 has been shown to interact with *H. pylori* during infection to block the bacteria from binding directly to the epithelial cells while activating the host inflammatory response. Furthermore, MUC1 in tumor cells has been shown to be aberrantly glycosylated, which results in overexpression of several novel tumor-associated carbohydrate structures that can serve as a possible prognostic marker or therapeutic target. A recent meta-analysis of ten translational studies demonstrate that MUC1 immunohistochemical expression is highly correlated with intestinal histology, vascular invasion, nodal metastasis, and decreased survival in gastric cancer patients [37].

Since AN people have a high prevalence of *H. pylori* infection, we evaluated the expression of MUC1 and the tumor-associated glycoform of MUC1 and detected the high expression of both in the majority of patients. Moreover, tumors that did not express TA-MUC1 were more likely to be at a lower stage. These results suggest MUC1 could be a relevant molecular biomarker and therapeutic target for AN gastric cancer patients that warrants further investigation. Similarly, mutations in *MUC16*, the gene that encodes the tumor antigen CA 125, have also recently been shown to correlate with tumor mutation load and improved clinical outcomes in GC patients [38].

The expression of COX-2 is elevated in the gastric mucosa of patients infected with *H. pylori* and has been shown to be prominent in premalignant and malignant lesions in the stomach. Similar to MUC1, COX-2 has been shown to stimulate innate immune responses and oncogenic signaling that drives gastric cancer initiation and progression [39]. There is a growing body of literature supporting the concept that COX-2 and MUC1 are components of a complex regulatory network that involves the WNT/CTNNB1 pathway [40]. This network functions as a driver pathway to promote tumorigenesis and metastasis [41] and is activated by carcinogenic *H. pylori* [42]. In support of this concept, a meta-analysis of 24 clinical studies reported regular use of non-steroidal anti-inflammatory drugs (NSAID) was inversely associated with gastric cancer risk [43]. Moreover, a recent randomized multi-center clinical trial demonstrated that addition of the COX-2 inhibitor celecoxib to chemotherapy improved progression-free survival and overall survival of gastric cancer patients whose tumors express COX-2 [44]. COX-2 is therefore, a potential therapeutic target for the prevention and treatment of gastric cancer. We observed that patients with high COX-2 expression were diagnosed at a later stage and had significantly reduced overall survival, suggesting that COX-2 promotes aggressive behavior of gastric cancers in AN people, consistent with what has been observed in other studies [45,46,47]. 

Many of the molecular factors investigated in this study are capable of altering the tumor microenvironment and can influence the initiation, progression, and therapeutic response of gastric cancer. Infectious agents, such as *H. pylori* and EBV, are significantly associated with gastric cancer and are both highly prevalent in AN patients [20]. Disruption of the immune equilibrium in the gastric mucosa, as is the case with *H. pylori* and EBV infection, may create a permissive immune microenvironment that allows gastric cancer cells to proliferate and migrate while evading immune detection [48,49,50]. EBV impacts the gastric tumor microenvironment by altering the immune-milieu [51], which in turn contributes to gastric cancer progression [52]. However, in EBVaGC the immune milieu and the presence of viral antigens are associated with improved patient survival [51] as well as targets for immune therapies [53]. Other factors, such as the upregulation of COX-2 and MUC1 in gastric cancer cells, have been shown to influence the microenvironment by permitting immune evasion [54,55], vessel invasion, and metastatic spread [56]. By targeting the COX-2 and MUC1, there is a potential to improve cancer immunogenicity [54,55].

Microsatellite instability (MSI) has been shown to be a predictive and prognostic biomarker in colorectal cancer but remains less established in gastric cancer. We, therefore, evaluated the MMR status of AN gastric cancers by assessing MSH2, MSH6, and PMS2 protein expression, which is lost in gastric cancers that are MSI-high. The incidence of MMR deficiency among AN patients is 10%, similar to Asians [57], and lower than the 22% that has been observed in Western populations, 22% [13,58,59]. A positive association was observed between MMR deficiency and patients who are female, over the age of 50, have an SNV mutation, and cancer in the distal stomach, consistent with the studies analyzing MMR deficiency with clinicopathological factors [60,61]. We, however, did not observe a correlation with subtype, which has been observed in some studies [57,62], but not others [58,59]. Furthermore, MMR-deficient patients had slightly poorer survival compared to MMR-stable patients, in particular MMR-deficient patients treated with perioperative chemotherapy (13.8 months, *n* = 4) compared to just surgery (19.3 months, *n* = 4). Our results are comparable to two clinical studies, MAGIC and CLASSIC trials, that observed MMR-deficient and MSI-high patients had inferior survival with perioperative chemotherapy [63,64].

Immune checkpoint blockade has been approved for the treatment of gastric cancer patients with locally advanced or metastatic gastric or gastroesophageal junction cancer in which there is >1% PD-L1 expressing cells. In our study, PD-L1 expression in immune and tumor cells was observed in 17% of AN gastric cancer patients. A PD-L1 combined positive score greater than 1 was significantly correlated with MMR deficient patients, similar to previous studies in breast [65], colorectal [66], gastric [67], and esophageal cancers [68,69]. Moreover, in gastric cancer, it has been well-known that PD-L1+ tumors frequently have *PIK3CA* mutations, which are also strongly associated with MSI-high and EBV+ tumors [13,59,70,71], as we observed in our cohort. *PIK3CA* inhibitors are now in clinical use in breast cancer and are being actively investigated in other tumor types [72]. Our work raises the possibility that PIK3CA mutations may also be a potential therapeutic target in AN gastric cancer. Additionally, cancers driven by the WNT/CTNNB1 pathway exhibit a unique immune signature that may be susceptible to inhibition by immune therapies [73]. Furthermore, it is reported that the overall clinical response rate of gastric cancer patients to PD-1 inhibition was significantly higher in patients with PD-L1+ tumor cells, *PIK3CA* mutation, MMR deficiency, and EBV+ tumors [53,74], suggesting these subsets could be candidates for immunotherapy. Further research into identifying prognostic biomarkers related to the tumor microenvironment may help identify novel molecular targets for improving treatment strategies in patients with gastric cancer.

## 4. Materials and Methods

### 4.1. Patient Samples

The use of archival tissue for research was reviewed and approved by the University of Alaska Anchorage Institutional Review Board (IRB), Mayo Clinic IRB, Alaska Area IRB, Southcentral Foundation Research Review Board, and the Alaska Native Tribal Health Consortium Health Research Review Committee (IRB protocol #665612). Formalin-fixed, paraffin-embedded (FFPE) archival cancer tissue biopsy and resection samples were collected from 85 AN patients with gastric cancer who underwent primary surgery between 2006 and 2014 at the Alaska Native Medical Center in Anchorage, AK. None of the patients had received pre- or perioperative neoadjuvant treatment. TNM-staging was performed according to the UICC/AJCC system 7th edition [75] and histopathological grading, according to the WHO [76]. The clinical and pathologic features of the case collections are given in Table 1. Follow-up data (overall survival) were available from all patients. 

### 4.2. Immunohistochemistry

FFPE tissue blocks were sectioned to 4 μm and mounted on Fisherbrand Superfrost Plus Microscope Slides (ThermoFisher), then dried at 60 °C for 1 h. Immunohistochemistry (IHC) staining was carried out on a Dako Autostainer Link 48 system (Agilent Technologies) using PD-L1 IHC 22C3 pharmDx kit (DAKO Agilent Technologies), COX-2 (clone CX-294 DAKO), MUC1 (clone E29 DAKO), and hypoglycosylated MUC1 (5E5, Creative Biolabs) with polymer refine detection (Leica DS9800), and counterstained with hematoxylin according to the manufacturer’s instructions.

### 4.3. Evaluation of Immunohistochemistry Staining

HER2 immunostaining was scored by one pathologist following a 4-step scale (0, 1+, 2+, 3+), according to the consensus panel recommendation on HER2 scoring for gastric cancer biopsies [77]. PD-L1, COX-2, and MUC1 expression were reviewed and enumerated independently and blindly by two experienced pathologists using the following quantification methods. *PD-L1 Quantification:* Tumor cell positivity was scored as the percentage of tumor cells exhibiting membrane staining of any intensity. Immune cell positivity was scored as the proportion of tumor area, including associated intratumor and contiguous peritumor stroma, occupied by PD-L1-stained immune cells at any intensity. Total cell positivity was scored as the percentage of positive cells, including tumor and immune/stromal cells, in all cells. The percentage of tumor cells showing positivity was recorded as less than or equal to 1%, and the percentage of immune/stromal cells showing positive staining was recorded as less than or equal to 1% and greater than 1%. A combined positive score that includes PD-L1 expressing cells (tumor and immune cells) was used for analysis [78]. All sections were reviewed. *COX-2 and MUC1 Quantification:* For each sample, at least five fields (inside the tumor and in the peripheral areas) were analyzed. Using a semi-quantitative scoring system microscopically and referring to each protein scoring method in other studies, stained tumor cells in each lesion were evaluated for protein expression intensity. For intensity assessment, staining intensity was scored as 0 (negative), 1 (weak), 2 (medium), and 3 (strong). Patients with a score ≤2 were grouped as low to moderate intensity, and patients with a score of >2 were grouped as high intensity.

### 4.4. EBV In Situ Hybridization

Freshly cut 4 μm slides were deparaffinized, and endogenous peroxidase activity was quenched by incubation in 1% H_2_O_2_ in methanol. Pretreatment with proteinase K for 20 min at room temperature (RT). Slides were washed in water then 95% ethanol and air-dried. The sections were incubated with an EBER probe (DAKO Cytomations, Glostrup, Denmark) for 90 min at 55 °C or with PBS for negative controls. Slides were immersed in Stringent Wash Solution (DAKO) for 25 min at 55 °C. Immunodetection was then performed with the anti-FITC/AP (DAKO) for 30 min at RT, then PNA substrate 60 min at RT. Slides were counterstained with hematoxylin (DAKO). For negative controls, the EBER probe was omitted. Cell analysis was performed by two independent investigators using light microscopy, at 40× or 20× magnification. A total of 10 representative microscopic fields were evaluated, and fields containing less than 5 cells were not considered. A gastric cancer sample positive for EBV was included as a positive control, and two slides treated without probe were used as negative controls. Samples, where 5% or more of the epithelial cells contained brown/red staining were considered positive. Although lymphocytes were also found to be infected by EBV, we did not include infected lymphoid cells in our analysis.

### 4.5. Mismatch Repair Status Determination

Tumor tissue mismatch repair (MMR) protein expression was determined by IHC in FFPE tissue sections. The monoclonal antibodies mouse anti-MSH2 (clone G219-1129, Ventana), mouse anti-MSH6 (clone 44, Ventana), rabbit anti-PSM2 (clone A16-4, Ventana) were used for IHC staining. MMR deficiency was determined when the tumor showed a loss or weak expression for the examined MMR proteins. Non-cancerous tissue adjacent to the tumor was used as a positive internal control. The evaluation was performed independently by two observers.

### 4.6. Solid Tumor-Targeted Cancer Gene Panel by Next-Generation Sequencing

FFPE tissue blocks were sectioned into eight 5-μm slides and manual microdissection to capture at least 6 mm^2^ area of tumor tissue. Samples were deparaffinized using QIAGEN’s deparaffinization solution following manufacturer’s guidelines. DNA extraction was performed using the QIAamp DNA FFPE tissue kit (QIAGEN). Extracted DNA was quantified using Qubit dsDNA BR Assay (ThermoFisher Scientific, Waltham, MA, USA). Next-generation sequencing was performed on the Illumina Miseq System using a commercial library kit (CAPN Illumina) to test for the presence of a mutation in targeted regions of the following 50 cancer-associated genes: *ABL1*, *AKT1*, *ALK*, *APC*, *ATM*, *BRAF*, *CDH1*, *CDKN2A*, *CSF1R*, *CTNNB1*, *EGFR*, *ERBB2*, *ERBB4*, *EZH2*, *FBXW7*, *FGFR1*, *FGFR2*, *FGFR3*, *FLT3*, *GNA11*, *GNAQ*, *GNAS*, *HNF1A*, *HRAS*, *IDH1*, *IDH2*, *JAK2*, *JAK3*, *KDR*, *KIT*, *KRAS*, *MET*, *MLH1*, *MPL*, *NOTCH1*, *NPM1*, *NRAS*, *PDGFRA*, *PIK3CA*, *PTEN*, *PTPN11*, *RB1*, *RET*, *SMAD4*, *SMARCB1*, *SMO*, *SRC*, *STK11*, *TP53,* and *VHL*. 

### 4.7. Statistical Analysis 

Association between various clinicopathological characteristics and overall survival were examined with Cox proportional hazard models. The Kaplan–Meier method was used for survival analysis, and differences in survival between groups were evaluated using the log-rank test. Variables with a *p* value < 0.05 on univariate analysis were included in the multivariate Cox proportional hazards regression model analysis. Correlation analyses were performed using the Kendall method (for non-parametric categorical and continuous variables). A multivariate Cox regression model was used to evaluate the prognostic significance of variables. *p*-values < 0.05 were considered statistically significant. All data were analyzed using IBM SPSS software (version 26 for Mac OS X, IBM, Armonk, NY, USA).

## 5. Conclusions

Gastric cancer is a leading cancer health disparity in the AN population. The most recent gastric cancer studies have expanded the classification of gastric cancer beyond tumor staging to include the various molecular features of this malignancy. However, these studies have not included AN people. Our study is the first to evaluate the molecular characteristics of gastric cancer in AN people, the population with the highest gastric cancer incidence in North America. Our findings suggest future avenues of investigation for clinical and translational studies. 

## Figures and Tables

**Figure 1 cancers-12-00198-f001:**
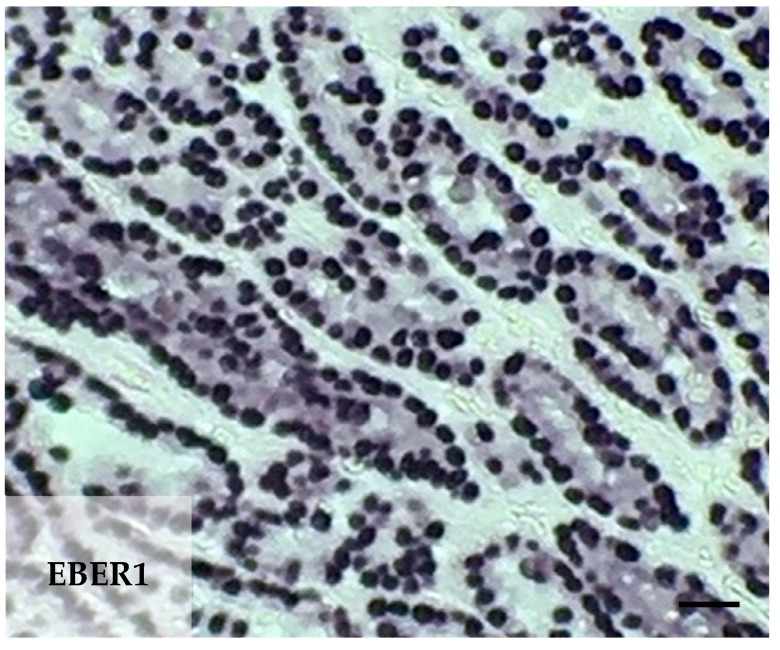
Epstein-Barr virus-associated gastric carcinoma. EBV-encoded small ribonucleic acid 1 (EBER1) in situ hybridization demonstrates positive nuclei in the carcinoma cells of an Alaska Native gastric cancer patient. Scale bar 25 μm.

**Figure 2 cancers-12-00198-f002:**
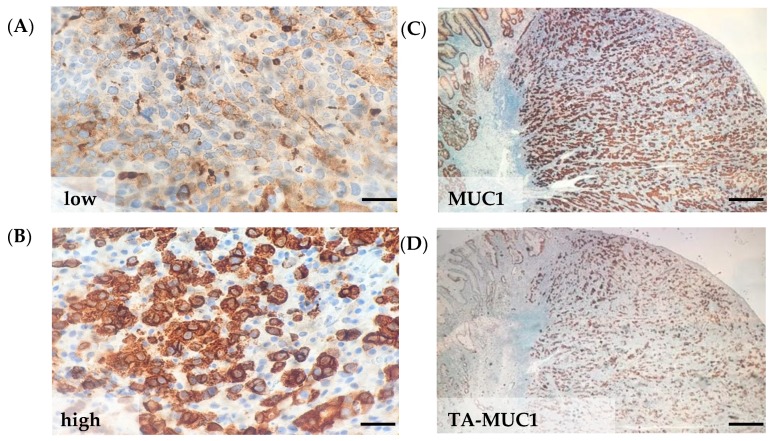
MUC1 and tumor-associated MUC1 protein expression in AN gastric cancer tissue. Representative MUC1 protein expression (**A**) low and (**B**) high in FFPE tumor tissue. Representative patient with (**C**) MUC1 and (**D**) tumor-associated MUC1 (TA-MUC1) expression. Scale bars; left panel, 25 μm and right panel 250 μm.

**Figure 3 cancers-12-00198-f003:**
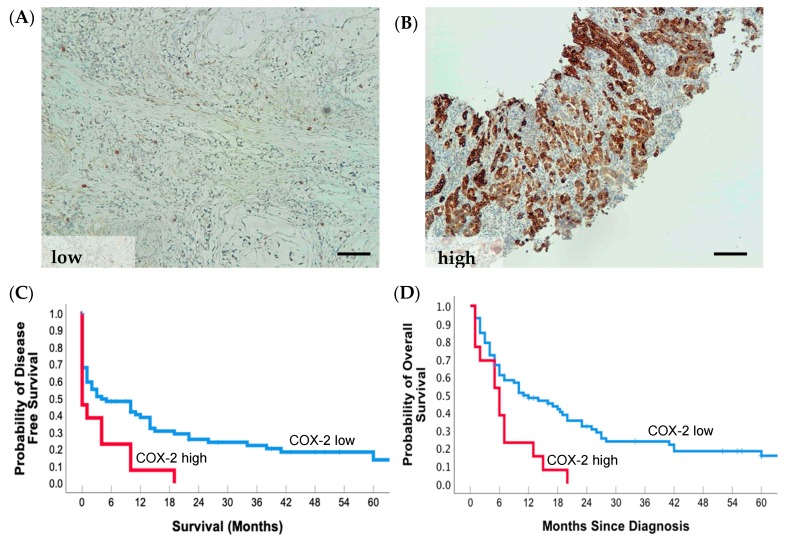
High COX-2 expression in gastric tumors correlates with reduced disease-free and overall survival in AN gastric cancer patients. Representative COX-2 protein expression (**A**) low and (**B**) high in gastric cancer tissue. (**C**) 5-year disease-free survival rate of patients with COX-2 high (*n* = 13) or low (*n* = 72) expressing tumors, *p* = 0.01. (**D**) 5-year overall survival rate of patients with COX-2 high (*n* = 13) or low (*n* = 72) expressing tumors, *p* = 0.004. Scale bars; 100 μm.

**Table 1 cancers-12-00198-t001:** Demographic, clinical, and pathological characteristics of Alaska Native (AN) gastric cancer molecular expression subgroup (*n* = 85) and next-generation sequencing (NGS) subgroup (*n* = 82).

Characteristics	Molecular Expression *n* (%)	NGS *n* (%)
Age		
Median (range)	60.9 (21–90)	61.2 (21–90)
Gender		
Male	52 (61.2)	51 (62.2)
Female	33 (38.8)	31 (37.8)
Stage at diagnosis		
I	23 (27.1)	18 (22.0)
II	16 (18.8)	17 (20.7)
III	12 (14.1)	11 (13.4)
IV	34 (40.0)	36 (43.9)
Anatomic site		
GE JX	13 (15.3)	14 (17.1)
Fundus	7 (8.2)	6 (7.3)
Body ^a^	21 (24.7)	20 (24.4)
Antrum	10 (11.8)	10 (12.2)
Pylorus	14 (16.5)	14 (17.1)
Overlapping	20 (23.5)	18 (22.0)
Lauren histological type		
Intestinal	41 (48.2)	41 (50.0)
Diffuse	44 (51.8)	41 (50.0)
Grade		
G2	25 (29.4)	25 (30.5)
G3	60 (70.6)	57 (69.5
Signet-ring cell presence		
No	60 (70.6)	60 (73.2)
Yes	25 (29.4)	22 (26.8)

^a^ Body includes lesser and greater curvature. Abbreviation: GE JX, gastroesophageal junction.

**Table 2 cancers-12-00198-t002:** Frequent single nucleotide variant (SNV) gene alterations.

Mutation Gene	Alaska Native *n* (%)	TCGA Nature 2014 *n* (%)
*TP53*	30 (36.6)	149 (46.8)
*PIK3CA*	14 (17.1)	72 (19.5)
*STK11*	6 (7.3)	3 (1.0)
*PTEN*	5 (6.1)	32 (7.5)
*KRAS*	5 (6.1)	28 (9.5)
*CTNN1B*	3 (3.7)	22 (6.1)
*APC*	3 (3.7)	53 (14.2)
*ERBB4*	3 (3.7)	47 (13.2)
*FBXW7*	2 (2.4)	28 (9.5)
*CDKN2A*	2 (2.4)	14 (4.7)
*SMAD4*	2 (2.4)	28 (9.5)
*EGFR*	1 (1.2)	15 (5.1)
*SMARCB1*	1 (1.2)	11 (3.7)
*GNAS*	1 (1.2)	19 (6.1)
*SMO*	1 (1.2)	14 (4.7)
*NRAS*	1 (1.2)	3 (1.0)
*JAK3*	1 (1.2)	13 (4.4)
*ATM*	1 (1.2)	41 (13.9)
*HRAS*	1 (1.2)	0 (0)
*CDH1*	1 (1.2)	25 (8.5)
*ERBB2*	1 (1.2)	15 (5.1)
*MET*	1 (1.2)	6 (2.0)
*RB1*	1 (1.2)	13 (4.4)
*BRAF*	1 (1.2)	16 (5.4)

Percentage (%) was calculated as the frequency of samples with said gene alteration (*n*) divided by the total of samples that passed DNA quality control (*n* = 82) for SNVs.

**Table 3 cancers-12-00198-t003:** Frequency of most prevalent single nucleotide variant (SNV) gene alterations.

Gene	Total Number of Samples *n*	82	295
	Function	Alaska Native *n* (%)	TCGA Nature 2014 *n* (%)
*TP53*			
R282W	Missense	3 (3.7)	5 (1.7)
R267W	Missense	2 (2.4)	0 (0)
G245S	Missense	2 (2.4)	2 (0.7)
H193R	Missense	1 (1.2)	2 (0.7)
R273H	Missense	1 (1.2)	10 (3.4)
R196*	Nonsense	1 (1.2)	3 (1.0)
*PIK3CA*			
E545K	Missense	7 (8.5)	12 (4.1)
H1047R	Missense	3 (3.7)	13 (4.4)
K111N	Missense	1 (1.2)	2 (0.7)
R88Q	Missense	1 (1.2)	4 (1.4)
*PTEN*			
S10N	Missense	1 (1.2)	0 (0)
R130Q	Missense	1 (1.2)	1 (0.3)
1026+2delT	IF deletion	1 (1.2)	0 (0)
Q214*	Truncating	1 (1.2)	0 (0)
*KRAS*			
G12D/A	Missense	2 (2.4)	12 (4.1)
G13D	Missense	1 (1.2)	10 (3.4)
A146T	Missense	1 (1.2)	2 (0.7)
*STK11*			
F354L	Missense	6 (7.3)	0 (0)
*CTNNB1*			
S37F	Missense	2 (2.4)	2 (0.7)
D32N	Missense	1 (1.2)	1 (0.3)
G34R	Missense	1 (1.2)	2 (0.7)

Abbreviations: IF, in frame. Percentage (%) was calculated as the frequency of samples with said gene alteration (*n*) divided by the total of samples that passed DNA quality control for SNVs.

**Table 4 cancers-12-00198-t004:** Immunohistochemistry tumor analysis (*n* = 85).

IHC Analysis	*n* (%)
PD-L1 CPS ^a^	
Negative, <1%	70 (82.4)
Positive, >1%	15 (17.6)
MMR deficient	
MSH-2	
Intact	84 (98.8)
Lost	1 (1.2)
MSH-6	
Intact	84 (98.8)
Lost	1 (1.2)
PMS-2	
Intact	78 (91.8)
Lost	7 (8.2)
HER2	
Negative	77 (90.6
Positive	8 (9.4)
CISH-EBER1	
Negative	66 (70.2)
Positive	19 (20.2)
COX-2 staining intensity	
Negative	4 (4.7)
Low to moderate	68 (80.0)
High	13 (15.3)
MUC1 staining intensity	
Low to moderate	15 (17.6)
High	70 (82.4)
TA-MUC1	
Negative	8 (9.4)
Positive	77 (90.6)

^a^ PDL-1 expression combined positive score (CPS) positive when >1% of tumor cells or >1% of immune cells. Abbreviations: *PD-L1* program death-ligand 1; *IHC* immunohistochemistry; *MMR* mismatch repair, *MSH-2* MutS protein homolog 2, *MSH-6* MutS protein homolog 6, *PMS-2* postmeiotic segregation increased 2; *HER2* human epidermal growth factor receptor 2; *CISH* chromogenic in situ hybridization; *EBER1* Epstein-Barr virus-encoded small ribonucleic acid 1; *COX-2* cyclooxygenase 2; *MUC1* mucin 1; *TA-MUC1* tumor-associated MUC1.

**Table 5 cancers-12-00198-t005:** Demographic, clinical, and pathological characteristics of AN EBVaGC subgroup (*n* = 19).

Characteristics	*n* (%)
Number of patients	19
Age	
Median	54
Range	32–80
Gender	
Male	13 (68.4)
Female	6 (31.6)
Stage at diagnosis	
I	4 (21.1)
II	4 (21.1)
III	2 (10.5)
IV	9 (47.4)
Location of primary tumor	
Distal esophagus GEJ	3 (15.8)
Fundus	3 (15.8)
Body	6 (31.6)
Antrum	1 (5.3)
Pylorus	1 (5.3)
Overlapping	5 (26.3)
Lauren histological type	
Intestinal	11 (57.9)
Diffuse	8 (42.1)
Grade	
G2	4 (21.1)
G3	15 (78.9)
Signet-ring cell presence	
No	15 (78.9)
Yes	4 (21.1)
PD-L1 CPS	
Negative, <1%	12 (63.2)
Positive, >1%	7 (36.8)
MMR deficient (MSI-high)	
MSH-2, MSH-6, PMS-2	
Intact	18 (94.7)
Lost	1 (5.3)
HER-2	
Negative	16 (84.2)
Positive	3 (15.8)
COX-2 staining intensity	
Negative	0 (0)
Low to moderate	19 (100)
High	0 (0)
MUC1 staining intensity	
Low to moderate	3 (15.8)
High	16 (84.2)
TA-MUC1	
Negative	1 (5.3)
Positive	18 (94.7)

Abbreviations: GE JX gastroesophageal junction I; *PD-L1* program death-ligand 1; *CPS* combined positive score; *IHC* immunohistochemistry; *MMR* mismatch repair, *MSH-2* MutS protein homolog 2, *MSH-6* MutS protein homolog 6, *PMS-2* postmeiotic segregation increased 2; *HER2* human epidermal growth factor receptor 2; *CISH* chromogenic in situ hybridization; *EBER1* Epstein-Barr virus-encoded small ribonucleic acid 1; *COX-2* cyclooxygenase 2; *MUC1* mucin 1; *TA-MUC1* tumor-associated MUC1.

**Table 6 cancers-12-00198-t006:** Univariate analyses for overall survival in Alaska Native gastric cancer patients.

Patient Characteristics	HR	95% CI	*P* ^a^
Age, >55 vs. <55 yr	0.60	0.38–0.96	0.031
Sex, male vs. female	1.13	0.72–1.79	0.601
Signet ring, absent vs. present	0.62	0.36–1.08	0.089
Lauren type, diffuse vs. intestinal	0.99	0.63–1.55	0.953
Grade, G2 vs. G3	1.67	0.92–3.03	0.089
Anatomic site			0.404
Cardia ^b^	1.00		
Noncardia ^c^	0.69	0.33–1.43	
Overlap/NOS	0.68	0.38–1.21	
AJCC Stage			**<0.0001**
I	1.00		
II	1.52	0.71–3.28	
III	1.90	0.82–4.40	
IV	6.16	3.11–12.20	
Treatment			**<0.0001**
Chemo	1.00		
None	2.19	1.22–3.95	
Neoadjuvant, surgery	0.20	0.09–0.43	
Surgery only	0.36	0.17–0.78	
Surgery, adjuvant	0.39	0.17–0.93	
Neoadjuvant, surgery, adjuvant	0.11	0.03–0.37	
Mutation, no vs. yes	1.41	0.80–2.48	0.232
*TP53* mutation, no vs. yes	1.05	0.64–1.72	0.857
*PIK3CA* mutation, no vs. yes	1.57	0.86–2.84	0.139
MMR, stable vs. deficient	1.40	0.67–2.93	0.374
COX-2, low vs. high	2.37	1.27–4.43	**0.007**
MUC1, low vs. high	1.24	0.65–2.37	0.518
TA-MUC1, low vs. high	1.42	0.84–2.38	0.188
HER2, negative vs. positive	1.13	0.88–1.45	0.322
PD-L1 CPS, <1 vs. >1	1.05	0.58–1.89	0.871
EBV, negative vs. positive	1.28	0.74–2.22	0.373

^a^ Bold type indicates statistical significance (*p* < 0.05). ^b^ Cardia includes gastroesophageal junction, fundus. ^c^ Noncardia includes, body, pylorus, and antrum. Abbreviations: HR, hazard ratio; CI, confidence interval; NOS, not otherwise specified; AJCC, American Joint Committee on Cancer; MMR, mismatch repair; *COX-2* cyclooxygenase 2; *MUC1* mucin 1; CPS, combined positive score. TA-MUC1, tumor-associated MUC1.

**Table 7 cancers-12-00198-t007:** Clinicopathologic factors predicating the prognosis in multivariate Cox regression models.

Patient Characteristics	HR	95% CI	*P* ^a^
Age, >55 vs. <55 yr	0.89	0.45–1.76	0.743
AJCC Stage			0.065
I	1.00		
II	1.05	0.42–2.60	
III	1.58	0.51–4.91	
IV	2.97	1.16–7.61	
Treatment			**0.002**
Chemo	1.00		
None	1.79	0.92–3.49	
Neoadjuvant, surgery	0.29	0.12–0.70	
Surgery only	0.60	0.22–1.64	
Surgery, adjuvant	0.51	0.15–1.73	
Neoadjuvant, surgery, adjuvant	0.12	0.02–0.59	
COX-2, low vs. high	0.89	0.45–1.76	0.743

^a^ Bold type indicates statistical significance (*p* < 0.05). Abbreviations: HR, Hazard ratio; CI, Confidence interval; AJCC, American Joint Committee on Cancer; *COX-2* cyclooxygenase 2.

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
