# Peer review of "Molecular Classification of Gastric Cancer among Alaska Native People"

_cancers, 2020, doi:10.3390/cancers12010198_

Round 1

Reviewer 1 Report

General comments:

The work is well developed and the description of the data is pleasant to read.

However, I add some comments that can improve the quality of this interesting work:

Methods: As a suggestion, because in gastric cancer, there´s clinical evidence of the utility of the combined positive score (CPS), for Pd-L1 expression, as a predictive biomarker of pembrolizumab,  it could be interesting to analyze the data accordingly. Results: The authors found an interesting result with 20% of patients EBV positive, more than expected compared with other populations around the world. It will be of great value if the authors could provide a more detailed description of the characteristics of this group: A table-specific with the characteristics of this subgroup could be useful including PD-L1 expression. In the discussion: the authors suggest that there is no robust data on the prognostic and predictive value of MSI response in gastric cancer. However, there´s evidence from the MAGIC trial published by Dr. Smyth and colleagues in JAMA 2017, and also from the CLASSIC TRIAL from Korea which showed that MSI patients do not benefit from chemotherapy at all. Moreover, due to the interesting results with 20% of patients EBV positive, the authors could value to provide a more detailed description of the study from Kim et al. Nature 2018 cited by the authors, and others as Derks et al. 2016,  regarding the  MSI / EBV subgroups.

Kind regards,

Author Response

 The work is well developed, and the description of the data is pleasant to read. However, I add some comments that can improve the quality of this interesting work:

Methods: As a suggestion, because in gastric cancer, there´s clinical evidence of the utility of the combined positive score (CPS), for Pd-L1 expression, as a predictive biomarker of pembrolizumab, it could be interesting to analyze the data accordingly.

Response to reviewer comment: We would like to thank the reviewer for their suggestion. We have reanalyzed the data using the combined positive score for PD-L1 expression.  We have included this data in the results, discussion and updated Table 4.

Reviewer One major comment: Results: The authors found an interesting result with 20% of patients EBV positive, more than expected compared with other populations around the world. It will be of great value if the authors could provide a more detailed description of the characteristics of this group: A table-specific with the characteristics of this subgroup could be useful including PD-L1 expression.

Response to reviewer comment:  We appreciate the reviewer’s suggestions. We have added a table to the results section that provides demographic, clinical, and pathological characteristics of the EBV positive patients, as shown below.

Table 5. Demographic, clinical, and pathological characteristics of AN EBVaGC subgroup (N=19)

Characteristics

n (%)

Number of patients

19

Age

   Median

54

   Range

32-80

Gender

   Male

13 (68.4)

   Female

6 (31.6)

Stage at diagnosis

   I

4 (21.1)

   II

4 (21.1)

   III

2 (10.5)

   IV

9 (47.4)

Location of primary tumor

   Distal esophagus GEJ

3 (15.8)

   Fundus

3 (15.8)

   Body

6 (31.6)

   Antrum

1 (5.3)

   Pylorus

1 (5.3)

   Overlapping

5 (26.3)

Lauren histological type

   Intestinal

11 (57.9)

   Diffuse

8 (42.1)

Grade

   G2

4 (21.1)

   G3

15 (78.9)

Signet-ring cell presence

   No

15 (78.9)

   Yes

4 (21.1)

PD-L1 CPS

   Negative or <1%

12 (63.2)

   Positive, >1%

7 (36.8)

MSI-H (MMR deficient)

   MSH-2, MSH-6, PMS-2

      Intact

18 (94.7)

      Lost

1 (5.3)

HER-2

   Negative

16 (84.2)

   Positive

3 (15.8)

COX-2 staining intensity

   Negative

0 (0)

   Low to moderate

19 (100)

   High

0 (0)

MUC1 staining intensity

   Low to moderate

3 (15.8)

   High

16 (84.2)

hMUC1

   Negative

1 (5.3)

   Positive

18 (94.7)

Abbreviations: GE JX, gastroesophageal junction I; PD-L1, program death-ligand 1; CPS, combined positive score; IHC immunohistochemistry; MMR mismatch repair, MSH-2 MutS protein homolog 2, MSH-6 MutS protein homolog 6, PMS-2 postmeiotic segregation increased 2; HER2 human epidermal growth factor receptor 2; CISH chromogenic in situ hybridization; EBER1 Epstein-Barr virus-encoded small ribonucleic acid 1; COX-2 cyclooxygenase 2; MUC1 mucin 1; hMUC1 hypoglycosylated MUC1.

Reviewer One major comment: In the discussion: the authors suggest that there is no robust data on the prognostic and predictive value of MSI response in gastric cancer. However, there´s evidence from the MAGIC trial published by Dr. Smyth and colleagues in JAMA 2017, and also from the CLASSIC TRIAL from Korea which showed that MSI patients do not benefit from chemotherapy at all.

Response to reviewer comment: Thank you for your comment. I have added the below text to the Discussion.

Furthermore, MMR deficient patients had slightly poorer survival (16.5 months) compared to MMR stable patients (24.6 months), in particular MMR deficient patients treated with perioperative chemotherapy (13.8 months). Our results are comparable to two clinical studies, MAGIC and CLASSIC trials, that observed MMR deficient and MSI-high patients had inferior survival with perioperative chemotherapy [41,42].

Reviewer One major comment: Moreover, due to the interesting results with 20% of patients EBV positive, the authors could value to provide a more detailed description of the study from Kim et al. Nature 2018 cited by the authors, and others as Derks et al. 2016, regarding the MSI / EBV subgroups.

Response: Thank you for your comment. I have added the below text to the Discussion.

Moreover, our observed association between PD-L1+ and EBV+ tumors has been reported in multiple studies [57,58]. Kim et al. demonstrated EBV+ tumors had superior response to PD-1 inhibitors and suggested EBV+ gastric cancer be actively considered for up-front pembrolizumab monotherapy similar to MSI-high tumors [57].

Reviewer 2 Report

Holly A. Martinson and colleagues provide interesting facts about the role of molecular classification and molecular markers in gastric cancer (GC) from eighty-five AN gastric adenocarcinoma patients using next-generation sequencing, immunohistochemistry, and in situ hybridization analysis. Overall, the authors provide original data, about a relatively “neglected” patients population. The study is interesting for the scientific community, however, there are some points that deserve to be improved before proceeding with the publication:

Do original clinical data exist about the translational relevance of the mentioned result? If yes, these elements should be presented, at least in the form of discussion and/or additional figure from a short literature meta-analysis, according to the authors preferences at least regarding MUC1 and COX2

2.6. COX-2 tumor cell expression and prognosis: Despite the functional role of COX-2 in human carcinogenesis, the relationship between COX-2 expression and the clinical prognosis of the diseases has not been clarified. Presently, Kaplan-Meier analysis showed the relationship between the expression of COX-2 and survival in GC patients with different clinicopathological factors (figure 3) OS was significantly negatively correlated with COX-2 expression in GC patients, do the authors checked for post-progression survival (PPS), and first progression (FP) or event/progression-free survival?

General comment: Moreover, the authors collected data from a retrospective cohort. Therefore, it would be useful to better clarify the methodological statistical approach used in order to limit and even avoid statistical biases. I would strongly recommend taking into account the interesting networks that can better explain the reported findings.

Firstly: a multivariate analysis can show the prognostic impact of several variables. Did the authors check for a statistical association between COX-2  and the other variables with significant impact within the uni- and multivariate analysis comprising additional factors that can impact on patients prognosis (i.e. TNM stages, therapy received, HER-2 status, etc.)

Those are fundamental details and information in this regard should be added to the results. In the frame of this thinking, and regarding the methods declared, I would point out that the biostatistical tests performed may be statistically significant but biologically less relevant if placed into a more complex context, such as a statistically powered prospective study. To compensate for these limits, the multivariate Cox’s proportional hazard regression models are a worthy tool. Nevertheless, a mandatory assumption needs to be taken into account in order to apply such a model: hazard proportionality. This assumption has to be made in order to proceed with the Cox model. If If the answer is affirmative, this should be better highlighted in materials and methods. If it is not, it is necessary to motivate and discuss the use of alternative models.

Discussion: One suggestion to take into account while discussing the data presented can be highlighting the role of the biomarkers identified in the light of more complex biological background. Indeed, MUC1 and COX-2 appear to be connected in a complex network (PMID: 24825509) involving WNT/CTNNB1 pathway, a well-known driver of cancer aggressiveness (PMID: 25992323) and tumour-dissemination (PMID: 31277479) with fundamental implication for immune-landscape, therapeutic approaches and patient molecular stratification. The authors should provide concise insights in this regard, with a particular biological focus discussing those network roles in aggressive phenotype acquisition.

Minor

Introduction/discussion: The authors mentioned that “Persistent inflammation during H. pylori or EBV infection can generate changes in the gut microenvironment where inflammatory mediators activate signaling pathways that lead to gastric tumorigenesis [reff 6,7].

The aforementioned literature and Medline review could be slightly improved, in order to extend the translational relevance of the exposed results. I'd suggest expanding. i.e. Has the bystander microenvironmental cells and precursors could be a bit better explained. I personally miss some important insights about tumor milieu role in mediating cancer progression, in both solid and hematologic tumors. Indeed, The cytokine- and cell-adhesion-dependent bone marrow niche and stromal microenvironment support new vessel formation and cancer proliferation, irrespective of immune-surveillance (PMID: 29628290). For instance, EBV appears to impact the immune-milieu and the patient's survival (PMID: 26498209), is implicated in the immune status equilibrium, a major driver of GC initiation (PMID: 29393912). This intimate interaction between GC cell, microenvironment via bystenders, i.e. endothelial cells, and CD8+ T cells creates a permissive immune microenvironment that allows undisturbed cancer proliferation (PMID: 16012717, PMID: 30546939).

I would recommend highlighting this topic, in order to better corroborate the translational relevance of the discussed data (i.e. MUC1 and COX2 role related to this immune-signatures)

Language: There are some linguistic gleanings that require a careful revision, a professional linguistic editing might be advisable.

Author Response

Thank you for your comments.

Reviewer Two General Comments: Holly A. Martinson and colleagues provide interesting facts about the role of molecular classification and molecular markers in gastric cancer (GC) from eighty-five AN gastric adenocarcinoma patients using next-generation sequencing, immunohistochemistry, and in situ hybridization analysis. Overall, the authors provide original data, about a relatively “neglected” patients population. The study is interesting for the scientific community, however, there are some points that deserve to be improved before proceeding with the publication:

Do original clinical data exist about the translational relevance of the mentioned result? If yes, these elements should be presented, at least in the form of discussion and/or additional figure from a short literature meta-analysis, according to the authors preferences at least regarding MUC1 and COX2.

Response to reviewer comment: Thank you for the comment. We have added a reference to a multi-center trial that demonstrated improved survival in gastric cancer patients with use of the COX-2 inhibitor celecoxib. We have also added text in support of the association of COX-2 and MUC1 with the WNT/CTNN1 pathway, as requested in the reviewer comment below.

Moreover, a recent randomized multi-center clinical trial demonstrated that addition of the COX-2 inhibitor celecoxib to chemotherapy improved progression-free survival and overall survival of gastric cancer patients whose tumors express COX-2 [42].

Reviewer Two major comment: 2.6. COX-2 tumor cell expression and prognosis: Despite the functional role of COX-2 in human carcinogenesis, the relationship between COX-2 expression and the clinical prognosis of the diseases has not been clarified. Presently, Kaplan-Meier analysis showed the relationship between the expression of COX-2 and survival in GC patients with different clinicopathological factors (figure 3) OS was significantly negatively correlated with COX-2 expression in GC patients, do the authors checked for post-progression survival (PPS), and first progression (FP) or event/progression-free survival?

Response to reviewer comment: We would like to thank the reviewer for their comment. We have added the disease-free survival curve to Figure 3 (please see attached file) to address the reviewers concerns. We also observed a significant difference in post-progression survival (3.2 vs 6.9 months, p=0.025) which we have included below  for the reviewer.

Reviewer Two major comment General comment: Moreover, the authors collected data from a retrospective cohort. Therefore, it would be useful to better clarify the methodological statistical approach used in order to limit and even avoid statistical biases. I would strongly recommend taking into account the interesting networks that can better explain the reported findings.

Firstly: a multivariate analysis can show the prognostic impact of several variables. Did the authors check for a statistical association between COX-2 and the other variables with significant impact within the uni- and multivariate analysis comprising additional factors that can impact on patients prognosis (i.e. TNM stages, therapy received, HER-2 status, etc.).

Those are fundamental details and information in this regard should be added to the results. In the frame of this thinking, and regarding the methods declared, I would point out that the biostatistical tests performed may be statistically significant but biologically less relevant if placed into a more complex context, such as a statistically powered prospective study. To compensate for these limits, the multivariate Cox’s proportional hazard regression models are a worthy tool. Nevertheless, a mandatory assumption needs to be taken into account in order to apply such a model: hazard proportionality. This assumption has to be made in order to proceed with the Cox model. If the answer is affirmative, this should be better highlighted in materials and methods. If it is not, it is necessary to motivate and discuss the use of alternative models.

Response to reviewer comment: We appreciate the reviewer’s comment. As suggested by the reviewer, univariate and multivariate analysis were performed to determine the prognostic impact of the study factors. Table 6 and 7 were added to the manuscript and are shown below.  As well as the following sentence to the results section 2.6 COX-2 tumor cell expression and prognosis.

Table 6. Univariate analyses for overall survival (OS) in Alaska Native gastric cancer patients

HR

95% CI

Pa

Age, >55 vs <55 yr

0.60

0.38-0.96

0.031

Sex, male vs female

1.13

0.72-1.79

0.601

Signet ring, absent vs present

0.62

0.36-1.08

0.089

Subtype, diffuse vs intestinal

0.99

0.63-1.55

0.953

Grade, G2 vs G3

1.67

0.92-3.03

0.089

Anatomic site

0.404

   Cardiab

1.00

   Noncardiac

0.69

0.33-1.43

   Overlap/NOS

0.68

0.38-1.21

AJCC Stage

<0.0001

   I

1.00

   II

1.52

0.71-3.28

   III

1.90

0.82-4.40

   IV

6.16

3.11-12.20

Treatment

<0.0001

   Chemo

1.00

   None

2.19

1.22-3.95

   Neoadjuvant, surgery

0.20

0.09-0.43

   Surgery only

0.36

0.17-0.78

   Surgery, adjuvant

0.39

0.17-0.93

   Neoadjuvant, surgery, adjuvant

0.11

0.03-0.37

Mutation, no vs yes

1.41

0.80-2.48

0.232

TP53 mutation, no vs yes

1.05

0.64-1.72

0.857

PIK3CA mutation, no vs yes

1.57

0.86-2.84

0.139

MMR, stable vs deficient

1.40

0.67-2.93

0.374

COX-2, low vs high

2.37

1.27-4.43

0.007

MUC1, low vs high

1.24

0.65-2.37

0.518

hMUC1, low vs high

1.42

0.84-2.38

0.188

HER2, negative vs positive

1.13

0.88-1.45

0.322

PD-L1 CPS, <1 vs >1

1.05

0.58-1.89

0.871

EBV, negative vs positive

1.28

0.74-2.22

0.373

aBold type indicates statistical significance (P<0.05). bCardia includes gastroesophageal junction, fundus. cNoncardia includes, body, pylorus, and antrum. Abbreviations: HR, Hazard ratio; CI, Confidence interval; NOS, not otherwise specified; AJCC, American Joint Committee on Cancer; MMR, mismatch repair; COX-2 cyclooxygenase 2; MUC1 mucin 1; CPS, combined positive score. hMUC1 hypoglycosylated MUC1.

Table 7. Clinicopathologic factors predicating the prognosis in multivariate Cox regression models

HR

95% CI

P

Age, >55 vs <55 yr

0.89

0.45-1.76

0.743

AJCC Stage

0.065

   I

1.00

   II

1.05

0.42-2.60

   III

1.58

0.51-4.91

   IV

2.97

1.16-7.61

Treatment

0.002

   Chemo

1.00

   None

1.79

0.92-3.49

   Neoadjuvant, surgery

0.29

0.12-0.70

   Surgery only

0.60

0.22-1.64

   Surgery, adjuvant

0.51

0.15-1.73

   Neoadjuvant, surgery, adjuvant

0.12

0.02-0.59

COX-2, low vs high

0.89

0.45-1.76

0.743

1Bold type indicates statistical significance (P<0.05). Abbreviations: HR, Hazard ratio; CI, Confidence interval; AJCC, American Joint Committee on Cancer; COX-2 cyclooxygenase 2.

After multivariate analysis, the difference in survival between COX-2 low and high was no longer significant (HR= 1.58, 95% CI = 0.80-3.13; P= 0.186; Table 7).

Reviewer Two major comment: Discussion: One suggestion to take into account while discussing the data presented can be highlighting the role of the biomarkers identified in the light of more complex biological background. Indeed, MUC1 and COX-2 appear to be connected in a complex network (PMID: 24825509) involving WNT/CTNNB1 pathway, a well-known driver of cancer aggressiveness (PMID: 25992323) and tumour-dissemination (PMID: 31277479) with fundamental implication for immune-landscape, therapeutic approaches and patient molecular stratification. The authors should provide concise insights in this regard, with a particular biological focus discussing those network roles in aggressive phenotype acquisition.

Response to reviewer comment: We would like to thank the reviewer for their comment. We have added additional discussion with regards to COX-2 and MUC1 and their connection to the WNT/CTNNB1 pathway.

There is a growing body of literature supporting the concept that COX-2 and MUC1 are components of a complex regulatory network that involves the WNT/CTNNB1 pathway [38].  This network functions as a driver pathway to promote tumorigenesis and metastasis [39] and is activated by carcinogenic H. pylori [40].

As well as the connection between WNT/CTNNB1 and immune signature susceptible to immune therapy (line 324).

Additionally, cancers driven by the WNT/CTNNB1 pathway exhibit a unique immune signature that may be susceptible to inhibition by immune therapies [40].

Reviewer Two major comment: Introduction/discussion: The authors mentioned that “Persistent inflammation during H. pylori or EBV infection can generate changes in the gut microenvironment where inflammatory mediators activate signaling pathways that lead to gastric tumorigenesis [reff 6,7].

Response to reviewer comment: Thank you for your comment. Additional references were added to this sentence.

Reviewer Two major comment: The aforementioned literature and Medline review could be slightly improved, in order to extend the translational relevance of the exposed results. I'd suggest expanding. i.e. Has the bystander microenvironmental cells and precursors could be a bit better explained. I personally miss some important insights about tumor milieu role in mediating cancer progression, in both solid and hematologic tumors. Indeed, The cytokine- and cell-adhesion-dependent bone marrow niche and stromal microenvironment support new vessel formation and cancer proliferation, irrespective of immune-surveillance (PMID: 29628290). For instance, EBV appears to impact the immune-milieu and the patient's survival (PMID: 26498209), is implicated in the immune status equilibrium, a major driver of GC initiation (PMID: 29393912). This intimate interaction between GC cell, microenvironment via bystenders, i.e. endothelial cells, and CD8+ T cells creates a permissive immune microenvironment that allows undisturbed cancer proliferation (PMID: 16012717, PMID: 30546939).

I would recommend highlighting this topic, in order to better corroborate the translational relevance of the discussed data (i.e. MUC1 and COX2 role related to this immune-signatures).

Response to reviewer comment: Thank you for the comment. We have added a paragraph in the discussion about these interactions.

Many of the molecular factors investigated in this study are capable of altering the tumor microenvironment and can influence the initiation, progression, and therapeutic response of gastric cancer. Infectious agents, such as H. pylori and EBV, are significantly associated with gastric cancer and are both highly prevalent in AN patients . Disruption of the immune equilibrium in the gastric mucosa, as is the case with H. pylori and EBV infection, creates a permissive immune microenvironment that allows gastric cancer cells to proliferate and migrate while evading immune detection [46-48]. EBV can impact the gastric tumor microenvironment by altering the immune-milieu [49], which in turn contributes to gastric cancer progression [50]. However, in EBVaGC the immune milieu and the presence of viral antigens have been shown to contribute to improved patient survival [49] as well as candidates for immune therapies [51]. Other factors, such as the upregulation of COX-2 and MUC1 in gastric cancer cells, has been shown to influence the microenvironment by permitting immune evasion [52,53], vessel invasion and metastatic spread (Tamura 2012. By targeting the COX-2 and MUC1, there is a potential to improve cancer immunogenicity [52,53]. Further research into identifying prognostic biomarkers related to the tumor microenvironment may help identify novel molecular targets for improving treatment strategies in patients with gastric cancer.

Reviewer Two comment: Language: There are some linguistic gleanings that require a careful revision, a professional linguistic editing might be advisable.

Response to reviewer comment: Thank you for your comment we have tried to carefully revise the manuscript. 

Reviewer 3 Report

In my opinion, the overall level of the paper is good structured: well designed and discussed, the procedures appear suitable and some important considerations are highlighted.

The conclusions sections provide useful information for the readers emphasizing the need of further clinical and translational studies, to better classify these tumors and to identify prognostic and predictive biomarkers for achieving the goal of precision therapy in GC patients.

Author Response

Response to Reviewer 3 comments: Thank you for your comments. 

Reviewer Three General Comments: In my opinion, the overall level of the paper is good structured: well designed and discussed, the procedures appear suitable and some important considerations are highlighted.

The conclusions sections provide useful information for the readers emphasizing the need of further clinical and translational studies, to better classify these tumors and to identify prognostic and predictive biomarkers for achieving the goal of precision therapy in GC patients.

Response to reviewer comments: We would like to thank the reviewer for taking the time to review our manuscript and providing us feedback.  

Round 2

Reviewer 2 Report

The authors have clarified several of the questions I raised in my previous review. Most of the major problems have been addressed by this revision. This means the strong conclusions put forward by this manuscript are warranted  and they are either novel and when not they are collectively definitive. I can approve the manuscript in this form.